# Optimal Selection and Experimental Verification of Wide-Bandgap Semiconductor for Betavoltaic Battery

**DOI:** 10.3390/nano15090635

**Published:** 2025-04-22

**Authors:** Jiachen Zhang, Kunlun Lv, Yuan Yin, Yuqian Gao, Ye Tian, Yuncheng Han, Jun Tang

**Affiliations:** 1School of Semiconductors and Physics, North University of China, Taiyuan 030051, China; hahaha.lv@foxmail.com (K.L.); 2201050217@st.nuc.edu.cn (Y.Y.); sz202419001@st.nuc.edu.cn (Y.G.); 20230127@st.nuc.edu.cn (Y.T.); tangjun@nuc.edu.cn (J.T.); 2Hefei Institutes of Physical Science, Chinese Academy of Sciences, Hefei 230031, China; yuncheng_han@163.com

**Keywords:** wide-bandgap semiconductor, betavoltaic battery, silicon carbide, gallium nitride, ultra-wide-bandgap semiconductor, EHP creation energy

## Abstract

Wide-bandgap semiconductor betavoltaic batteries have a promising prospect in Micro-Electro-Mechanical Systems for high power density and long working life, but their material selection is still controversial. Specifically, the silicon carbide (SiC) betavoltaic battery was reported to have higher efficiency, although its bandgap is lower than that of gallium nitride (GaN) or diamond, which is inconsistent with general assumptions. In this work, the effects of different semiconductor characteristics on the battery energy conversion process are systematically analyzed to explain this phenomenon, including beta particle energy deposition, electron–hole pair (EHP) creation energy and EHPs collection efficiency. Device efficiencies of the betavoltaic battery using SiC, GaN, diamond, gallium oxide (Ga_2_O_3_), aluminum nitride (AlN) and boron nitride (BN) are compared to determine the optimum semiconductor. Results show that SiC for the betavoltaic battery has higher efficiency than GaN, Ga_2_O_3_ and AlN because of higher EHPs collection efficiency, less energy loss and fewer material defects, which is the optimal selection currently. SiC betavoltaic batteries were prepared, with the device efficiency having reached 14.88% under an electron beam, and the device efficiency recorded as 7.31% under an isotope source, which are consistent with the predicted results. This work provides a theoretical and experimental foundation for the material selection of betavoltaic batteries.

## 1. Introduction

With the rapid development of micro-electro-mechanical systems (MEMS), micro-energy generators are especially in demand in extreme environmental conditions, such as deep space, the deep sea and the polar region, where there are extreme ambient temperatures, weak sunlight intensity and high maintenance costs [1]. Conventional micro-electrochemical batteries require frequent charging and degrade easily in extreme environments, which limits their application in MEMS [2]. The betavoltaic battery is an energy generator converting the decay energy of isotope sources into electricity, which has significant potential as an alternative energy generator due to its long operational life, high energy density, easy miniaturization and strong resistance to extreme environments.

A betavoltaic battery mainly consists of an isotope source and a semiconductor energy converter [3]. The working principle of the betavoltaic battery is shown in Figure 1a, which consists of the following steps: beta particles decay from the isotope source, with part of them emitting from the source surface, passing through the electrode and injecting into the semiconductor material; the electron–hole pairs (EHPs) are created and then separated by the built-in electric field of the semiconductor junction, such as the Schottky or PN junction; and, finally, electrons and holes are collected by electrodes to generate currents. During the transportation processes of beta particles, self-absorption of the isotope source, blocking of the electrode metal and backscattering of the semiconductor material lead to energy loss, which makes no contribution to the battery output performance. In processes of EHPs separation and collection in semiconductors, the EHPs recombination and electrode resistance will further reduce the output performance. In addition, during the generation of EHPs, the EHP creation energy of the semiconductor will also affect the output performance of the battery. The higher EHP creation energy usually leads to a smaller radiation generation current because of fewer EHPs created.

The total energy conversion efficiency (*η*) refers to the proportion of battery output power to isotope source power, which can be divided into isotope source efficiency (*η_s_*) and semiconductor device efficiency (*η_d_*). The *η_s_* is inversely proportional to the thickness of the isotope source. A thinner isotope source tends to have a higher source efficiency and results in a lower battery power density at the same time. To determine the optimum thickness of the isotope source, the *η_s_* and power density of the betavoltaic battery should be balanced to obtain a high output power [4]. ^63^Ni is the most common isotope source for betavoltaic batteries due to its long half-life of 100 years and soft beta particle energy below the irradiation damage threshold of the semiconductor lattice, which is about 200 keV [5]. The corresponding *η_s_* is about 35% when the thickness of ^63^Ni is 500 nm with single-side emission. An electron beam from a scanning electron microscope is often used to simulate isotope sources because of its controllable and accurate location, direction and energy.

**Figure 1 nanomaterials-15-00635-f001:**
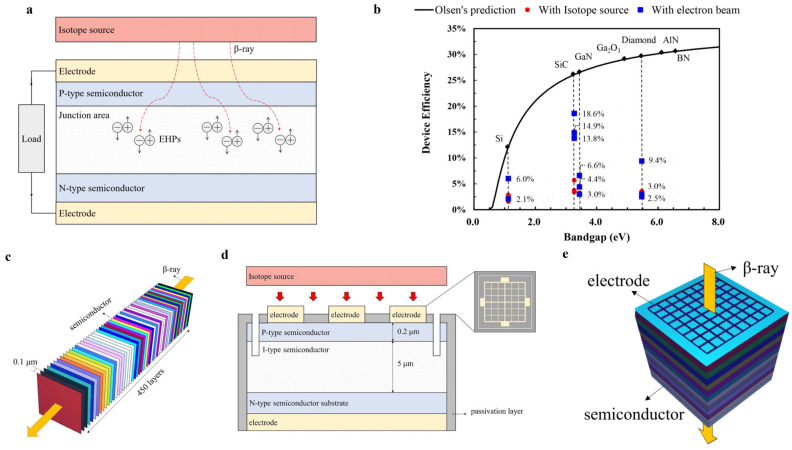
Models of the device. (**a**) Typical principles of a betavoltaic battery with PN junction. (**b**) Larry Olsen’s prediction and the device efficiency of a betavoltaic battery experimental device [6,7,8,9,10,11,12,13,14,15]. (**c**) The ideal simulation model of the betavoltaic battery; the colors in the figure are used to highlight the layering. (**d**) The prediction model of the betavoltaic battery experimental device. (**e**) Simulation model reflecting actual experimental conditions.

The *η_d_* is closely related to the characteristics of the used semiconductor material. Larry Olsen [3] proposed that *η_d_* is positively correlated with the semiconductor bandgap, i.e., betavoltaic batteries with wider bandgap semiconductors have higher efficiency, as shown in Figure 1b. Typical wide-bandgap semiconductors used for betavoltaic batteries include silicon carbide (SiC), gallium nitride (GaN) and diamond, with bandgaps as 3.27 eV, 3.44 eV and 5.47 eV, respectively. However, the selection of optimal wide-bandgap semiconductors for high-efficiency betavoltaic batteries is still controversial. Chen Zhao et al. [6] considered SiC as the optimum energy converter for betavoltaic batteries because it has a low recombination current. Szymon Grzanka et al. [7] chose GaN as the semiconductor material for a betavoltaic battery due to its high material density, which increases the absorption of beta particles. C. Delfaure et al. [8] selected diamond as the energy converter for the reasons of the low electron backscattering and the high diffusion length of EHPs. GaN and diamond have been widely studied in betavoltaic batteries in recent years due to their large bandgap and strong irradiation resistance, with their maximum experimental achieved *η_d_* as 6.6% [9] and 9.4% [8], respectively, as shown in Figure 1b, while the experimental *η_d_* of SiC betavoltaic batteries were already more than 12% [6,10,11], being significantly higher than those of GaN and diamond betavoltaic batteries, which is inconsistent with Larry Olsen’s prediction. We need further systematic analysis to obtain a better efficiency performance of betavoltaic batteries.

Recently developed ultra-wide-bandgap semiconductors (*E_g_* > 4 eV), such as gallium oxide (Ga_2_O_3_), aluminum nitride (AlN) and boron nitride (BN), have promising applications in betavoltaic batteries in the future due to their large bandgap, strong irradiation resistance and low intrinsic carrier concentration. In addition, combined with innovations in thermoelectric and piezoelectric materials, these semiconductors show great potential for energy recovery, self-powered sensors and more [16,17]. Although the material fabrication technologies of ultra-wide-bandgap semiconductors are not mature enough to prepare stable devices [18], the theoretical study of betavoltaic batteries with ultra-wide-bandgap semiconductors is worth anticipating.

The purpose of this work is to establish the optimal selection principles for the wide-bandgap semiconductors applied in betavoltaic batteries through systematical analysis and experimental verification. By analyzing principal factors affecting the EHPs generation and transport processes, including beta particle energy deposition, EHP creation energy, and EHPs collection efficiency, the *η_d_* of betavoltaic batteries using different semiconductors (SiC, GaN and diamond) were compared by empirical formula calculation and Monte Carlo simulation. Furthermore, the theoretical *η_d_* of betavoltaic batteries with ultra-wide-bandgap semiconductors were predicted. The optimal wide-bandgap semiconductor betavoltaic battery device is designed and fabricated experimentally, and its output current–voltage (I–V) characteristics were measured and compared with those of betavoltaic batteries from other semiconductor experiments.

## 2. Methods

### 2.1. Theoretical Model

#### 2.1.1. Efficiency Calculation

The total efficiency *η* is one of the principal factors for evaluating the performance of the betavoltaic battery, which can be described as(1)$$\begin{array}{c} \eta =\frac{P_{max}}{P_{in}}=\frac{I_{sc}\cdot V_{oc}\cdot FF}{A\cdot E_{ave}\cdot q} \end{array}$$
where *P_max_* is the maximum output power of the battery; *P_in_* is the power of the isotope source; *I_sc_* is the short-circuit current; *V_oc_* is the open-circuit voltage; *FF* is the filling factor; *A* is the activity of the isotope source; *E_ave_* is the average energy of isotope source charged particles; and *q* is the electron charge.

*V_oc_*, *I_sc_* and *FF* can be obtained by analyzing the I–V characteristics of betavoltaic battery experimental devices and simulating beta particles’ energy deposition combined with empirical formulas. *I_sc_* can be expressed as in [19].(2)$$\begin{array}{c} I_{sc}=\int_{0}^{H}q\cdot G\left(x\right)\cdot CE\left(x\right)dx \end{array}$$
where *H* is the thickness of the semiconductor; *x* is the distance between the energy deposition location of the incident particle and the semiconductor surface; *G(x)* is the production rate of EHPs; and *CE(x)* is the collection rate of EHPs.

*G(x)* is closely related to the energy deposition of beta particles and can be expressed as(3)$$G\left(x\right)=\frac{A\cdot E\left(x\right)}{\varepsilon }$$
where *ε* is the EHP creation energy, which can represent the energy to generate one single EHP; and *E(x)* is the energy deposition function.

*CE(x)* is related to the position of EHPs generation and the semiconductor junction region and can be expressed as in [20].(4)$$\begin{array}{c} CE\left(x\right)=\left\{\begin{array}{c} 1-tanh\frac{x}{L_{p}},\;x>0\;\;\;\\ 1,\;\;\;\;x\leq 0 \end{array}\right. \end{array}$$
where *L_p_* is the diffusion length of minority carriers, which is closely related to material defects; and *x* is the distance between the EHPs generation position and the junction region, which is related to the width of junction region *W*. When *x* ≤ 0, EHPs are generated in the junction region and can be collected completely (i.e., *CE(x)* is 100%).

The width of the PN junction region is affected by the intrinsic carrier concentration *n_i_* and the doping concentration of the semiconductor and can be expressed as follows, as in [21].(5)$$\begin{array}{c} W=\sqrt{V_{bi}\left(\frac{2\varepsilon_{r}\varepsilon_{0}}{q}\right)\left(\frac{N_{A}+N_{D}}{N_{A}N_{D}}\right)}=\sqrt{\frac{2kT\varepsilon_{0}\varepsilon_{r}}{q^{2}}\cdot \ln\left(\frac{N_{A}N_{D}}{{n_{i}}^{2}}\right)\cdot \left(\frac{N_{A}+N_{D}}{N_{A}N_{D}}\right)} \end{array}$$
where *V_bi_* is the built-in potential of the PN junction; *ε*_0_ is the vacuum permittivity; *ε_r_* is the relative permittivity of semiconductor; *N_A_* is the P-type semiconductor doping concentration; *N_D_* is the N-type semiconductor doping concentration; *k* is the Boltzmann constant; and *T* is the working temperature.

Considering that the recombination current affects the I–V characteristics, the *V_oc_* can be expressed as in [10].(6)$$\begin{array}{c} V_{oc}=\frac{nkT}{q}ln\left(\frac{I_{sc}}{I_{0}}+1\right) \end{array}$$
where *n* is the ideality factor, which reflects the proportional relationship between the diffusion current and the recombination current in PN junction, generally ranging from 1 to 2 [22]; and *I*_0_ is the proportional coefficient, which is related to the diffusion coefficient, minority lifetime and diffusion length of minority carriers [10].

Without considering the influence of series and parallel resistances, *FF* can be derived from the Shockley equation and expressed as:(7)$$\begin{array}{c} FF≅\left\{1-\frac{1}{ln\left(\frac{I_{SC}}{I_{0}}\right)}\right\}\left\{1-\frac{ln\left[ln\left(\frac{I_{SC}}{I_{0}}\right)\right]}{ln\left(\frac{I_{SC}}{I_{0}}\right)}\right\} \end{array}$$

#### 2.1.2. Upper-Limit Efficiency

Larry Olsen assumed that *CE(x)* in the semiconductor is 100%, and *I*_0_ only considers radiation recombination; the upper-limit device efficiency can be deduced by Formulas (1), (2) and (6), as in [3].(8)$$\begin{array}{c} \eta_{d}=\frac{V_{oc}\cdot FF}{\varepsilon } \end{array}$$
where *ε* was described by Shockley and Klein based on experimental results, consisting of bandgap losses *E_g_*, thermalization losses *E_f_*, and optical phonon interactions *γE_γ_*. According to the simple two-band (STB) approximation, where *E_f_* = 0.9 *E_g_*, *ε* is approximately as in [3,20,23].(9)$$\begin{array}{c} \varepsilon =E_{g}+2E_{f}+\gamma E_{\gamma }=2.8E_{g}+{\gamma E}_{\gamma }\approx 2.8E_{g}+0.5\;eV \end{array}$$

The relationship between the EHP creation energy and bandgap in Equation (9) is also known as the Shockley–Klein relationship.

#### 2.1.3. Energy Deposition Simulation

The performance of the betavoltaic battery with different semiconductors can be predicted based on Formulas (1)–(7) by combination with Monte Carlo simulations.

The ideal theoretical model consists of an isotope source and a semiconductor. The semiconductor includes P-type and I-type regions, and the doping concentrations were 1 × 10^19^ cm^−3^ and 1 × 10^15^ cm^−3^, respectively. The junction region widths of each semiconductor were 1.80 μm (SiC), 1.94 μm (GaN) and 1.82 μm (diamond) at this doping concentration, according to Equation (5). The semiconductor was segmented into 450 layers, with a minimum thickness of 0.1 μm to ensure the simulation accuracy, as shown in Figure 1c. The isotope source was set as one side-emitting surface ^63^Ni source, and the source efficiency of the mono-energy electron beam was assumed as 100%.

The prediction model of the experimental device consists of electrodes, a passivation layer, a 0.2 μm P-type semiconductor region and a 5 μm I-type semiconductor region, as shown in Figure 1d. The isotope source was a volume source, with consideration of the self-absorption effect, and the electron beam was a unidirectional-emitting single-energy surface source with a beam intensity of 90 pA (equivalent to 15 mCi). The recombination effects associated, because of defects introduced during fabrication, were not considered, and the ideal factor of the experimental device was set to 1.

The controversy mentioned in Section 1 can be theoretically analyzed by using the theoretical model mentioned above to determine the current optimal selection of wide-bandgap semiconductor for the betavoltaic battery. The output performance of a betavoltaic battery with the selected semiconductor will be further verified by the following experiments.

### 2.2. Experiment

#### 2.2.1. Fabrication

As demonstrated in Figure 1d, the SiC-based betavoltaic battery device conceptualized in this study incorporates semiconductors with PIN structures, surface passivation and electrodes. The N-type semiconductor substrate has a resistivity of 0.017 Ω·cm with a thickness of 365.5 μm. The P-type and I-type structures of the semiconductors were fabricated by chemical vapor deposition, corresponding to lightly doped N-type layers and heavily doped P-type layers, respectively. The N-type layer exhibits a doping concentration of 1 × 10^15^ cm^−3^ and a thickness of 5 μm, while the P-type layer demonstrates a doping concentration of 1 × 10^19^ cm^−3^ and a thickness of 0.2 μm. The effective area of the prepared semiconductor transducer device is 4 mm × 4 mm. The surface passivation was performed by dry oxygen oxidation for 3 h at 1100 °C and annealing in N_2_ with constant temperature for 1 h. The passivation layer covered all surfaces of the PIN structure except the electrode to eliminate the surface state and reduce leakage current, with a thickness of about 24 nm. The electrodes were deposited by magnetron sputtering with a multilayer structure and subjected to rapid thermal annealing (RTA) at 1050 °C for 3 min in N_2_ atmosphere to form Ohmic contact. In the present study, Ni/Ti/Al/Ni multilayers were selected as electrodes on the P-type semiconductor surface, patterned in a grid-like pattern with 9% of the total active area in order to minimize the loss of penetration energy of beta particles, with a thickness of less than 200 nm. Similarly, Ni/Ti/Al multilayers were selected as electrodes on the N-type semiconductor surface, and the electrodes on the N-type substrate were completely covered in order to ensure low electrode resistance.

#### 2.2.2. Measurement

I–V characteristics of the devices were maintained both under the condition of isotope source and of electron beam. A Keithley 4200A-SCS semiconductor parameter analyzer (Tektronix, Beaverton, OR, USA)obtained I–V characteristics of devices under an isotope source, with the devices placed in a Faraday shielding chamber to prevent the effects of natural light and electromagnetic interference. The ^63^Ni isotope source has an active area of 1 × 3 cm^2^ with total activity as 15 mCi, and the physical diagram of the devices is shown in Figure 2. The electron beam simulation measurement system was built by the China Academy of Engineering Physics and consisted of a scanning electron microscope (KYKY-EM6200, KYKY Technology Co., Ltd., Beijing, China) to generate electron beams with different kinetic energy, a Faraday Cup to measure the electron beam current generated by the electron gun and a Keithley 6487 picoammeter (Tektronix, Beaverton, OR, USA) to measure the I–V characteristics of the devices [24]. The acceleration voltage of the electron beam is 20 kV with a fixed beam intensity of 90 pA, which simulates the energy of emitted beta particles of the ^63^Ni isotope source.

## 3. Results and Discussion

### 3.1. Selection of Typical Wide-Bandgap Semiconductor

#### 3.1.1. Effect of the Electron–Hole Pair Creation Energy

Assuming that the collection rate of EHPs is 100% and the energy of beta particles is all deposited in the semiconductor, the upper-limit device efficiency of the betavoltaic battery is mainly affected by the EHP creation energy. In Olsen’s prediction, the EHP creation energy followed the Shockley–Klein relationship with the bandgap, and the upper-limit device efficiency is shown as the black line in Figure 3a, according to Formulas (2) and (6)–(9). It can be seen that the upper-limit device efficiency increases positively with the semiconductor bandgap, reaching to a maximum of about 30%.

However, the validity of the Shockley–Klein relationship between the *ε* and the bandgap in Equation (9) has been widely questioned [25,26,27,28,29]. Because it is based on the early semiconductor’s experimental results, where the semiconductor materials had many defects and low carrier collection efficiencies, the *ε* predicted by the Shockley–Klein relationship produced high apparent values [30]. In addition, the complex bandgap structures of wide-bandgap semiconductors cause inter-band tunneling and make the STB approximation not as accurate as classical semiconductors with a narrow bandgap, which violates the hypothesis described as Equation (9), and the *ε* will be lower than the Shockley–Klein relationship’s prediction [22]. According to the experimental results of the *ε*, many studies have modified the Shockley–Klein relationship and re-fitted various relationships, as shown in Figure 3b [25,26,27,28,29]. However, these relationships still cannot describe the relationship between the *ε* experimental value and the bandgap well, which will lead to an underestimation of the theoretical efficiency of the betavoltaic battery [25].

The red dots in Figure 3b show the currently available experimental *ε* values of SiC, GaN and diamond [22,23,25,26,27,28,29,30,31,32,33,34,35,36]. Many studies have been conducted on the EHP generation energy of SiC, because SiC is a good candidate for high-temperature alpha detectors due to its wide bandgap and low noise. Meanwhile, the EHP creation energy of GaN has been less studied and is more advantageous in X-ray detectors due to its large atomic number. The experimental *ε* of GaN is closer to the Shockley–Klein relationship than that of SiC and diamond, as shown in Figure 3b, which means its ratio to the bandgap is highest, and the upper-limit efficiency of a GaN betavoltaic battery will be the lowest according to Equation (8). The modified upper-limit device efficiency is shown as the red line in Figure 3a based on experimental values *ε′*, where the error lines around the red dots represent the results for different *ε′*. As can be seen from Figure 3a, the upper-limit device efficiency is no longer positively correlated with the bandgap and is higher than Larry Olsen’s prediction, since the *ε′* and the bandgap do not satisfy the Shockley–Klein relationship.

**Figure 3 nanomaterials-15-00635-f003:**
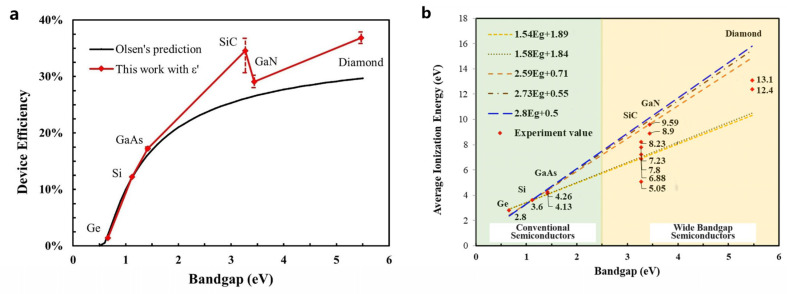
Effect of the electron-hole pair creation energy. (**a**) The upper-limit device efficiencies of a betavoltaic battery with different semiconductors based on the Shockley-Klein relationship of ε (Olsen’s prediction) and experimental ε′ values (this work). (**b**) Experimental and theoretical values of EHP creation energy for different semiconductors [20,23,25,26,27,28,30,32,33,36].

#### 3.1.2. Effect of the Energy Deposition of Beta Particles

The energy of beta particles is not all deposited in the semiconductor, as in Olsen’s prediction, but a fraction of electrons is returned from the semiconductor surface by elastic scattering, called backscattering, and its amount is closely related to the nucleus size of the semiconductor. The interaction process and transport path of beta particles and different semiconductors are simulated through the Geant4 program, as shown in Figure 4a. In this simulation, the beta particle sources are set as a point source to clearly display the physical process. It can be seen that GaN has the greatest backscattering caused by the highest density and largest atomic number compared with those of SiC and diamond.

The distribution of the energy deposition of beta particles in semiconductors is also important for the battery’s efficiency, which affects the EHPs generation process, as shown in Equations (2) and (3). The energy deposition distribution of beta particles in each semiconductor was simulated by a continuous-spectrum one-side-emitting surface ^63^Ni source, as shown in Figure 4b. Due to the large atomic number, the energy of beta particles deposited in GaN is mainly closer to the incident surface than that of SiC and diamond, resulting in more energy deposited in the P-type layer. In the PIN structure device, the junction region mainly falls in the N^−^-type layer with a low doping concentration, while it is thin in the P-type layer with ahigh doping concentration. The width of the depletion region in the P-type layer is only 18.0 nm (SiC), 19.4 nm (GaN) and 18.2 nm (diamond) at the doping concentration in Section 2.1.3, and the energy deposited in a P-type layer is not conducive to EHPs collection. Figure 4b shows the proportion of the energy deposition of beta particles in SiC, GaN and diamond at different depths to the total energy of the ^63^Ni source. The total deposition energy of beta particles in GaN is 8% lower than that of SiC and 13% lower than that of diamond due to the previously mentioned backscattering, and the energy deposited in the first 5 μm of the junction region is also lower than that of SiC and diamond. These phenomena indicate that there is more energy loss in GaN, which has a negative impact on the battery’s energy conversion efficiency.

#### 3.1.3. Effect of the Electron–Hole Pairs Collection Efficiency

The EHP collection process has a significant impact on the efficiency of the battery [37,38]. The *CE(x)* outside the semiconductor junction region is actually less than 100%, and it is determined by the junction width *W* and the minority carrier diffusion length *L_p_*, as in Equation (4). As mentioned in Section 2.1.3, there is no obvious difference in the junction widths for SiC, GaN and diamond under the same doping concentration conditions, which means that *L_p_* significantly affects *CE(x)*. The *L_p_* is closely related to the semiconductor band structure, impurity content and surface state, and it is affected by direct recombination, surface recombination, interface recombination and material defects [22]. GaN is a direct bandgap semiconductor, and its *L_p_* is shorter than that of SiC, diamond and other indirect bandgap semiconductors [39]. Moreover, different from the homogeneous epitaxy of SiC and diamond, GaN is generally prepared by heteroepitaxial growth due to technical limitations. The lattice mismatch between the GaN epitaxy material and the substrate will introduce extra defects [40], which greatly reduces the *L_p_* of GaN. The *L_p_* of different semiconductors are shown in Figure 5a. The *L_p_* of GaN is 1.9 μm, which is the smallest among the listed semiconductor materials. According to Equation (4), the minimum *L_p_* of GaN will result in its *CE(x)* outside the junction region being lower than that of other semiconductors. The *CE(x)* of different semiconductors is shown in Figure 5a, which shows the *CE(x)* of GaN significantly decreases outside the junction region. The *CE(x)* of GaN has been reduced to 35% at 5 μm from the junction edge, and almost no charge carriers are collected beyond 12 μm. In contrast, the *CE(x)* of SiC is significantly higher than that of GaN, with 62% at 5 μm and 14% at 12 μm, and the *CE(x)* of diamond is higher, with 72% at 5 μm and 26% still at 12 μm. The higher collection efficiency of SiC and diamond than GaN ensures that the EHPs generated outside the junction region can be collected more, which will be beneficial to obtain higher energy conversion efficiency.

#### 3.1.4. The Ideal Efficiency

The radiation-generated current distribution of SiC, GaN and diamond betavoltaic batteries can be obtained by considering the energy deposition distribution, average ionization energy and collection efficiency, as shown in Figure 5b. It can be seen that the short-circuit current of GaN is larger than that of diamond and SiC in the first 1 μm and then lower after 1 μm. This is due to the high atomic number and density of GaN, which makes the energy deposition distribution of beta particles in GaN shallow, leading to the radiation-generated current of its battery being mostly concentrated near the surface. The ideal efficiency of a betavoltaic battery using SiC, GaN or diamond is shown in Table 1, without considering the effects of isotope source self-absorption, device electrodes and passivation. The large bandgap of semiconductors will make it difficult for electrons to transition from the valence band to the conduction band, resulting in a low intrinsic carrier concentration. According to Equation (6), the lower the intrinsic carrier concentration of the semiconductor, the higher the PN junction’s built-in electric field, with the same condition of doping concentrations. This means that the betavoltaic battery using semiconductors with a larger bandgap has a higher *V_oc_*, and the *V_oc_* of the GaN betavoltaic battery tends to be actually higher than that of the SiC betavoltaic battery. However, due to the smaller collection efficiency and higher energy loss, the GaN betavoltaic battery has about 20% lower radiation-generated current than the SiC betavoltaic battery, resulting in its ideal efficiency being the lowest among the three kinds of materials. Generally speaking, a betavoltaic battery using a wide-bandgap semiconductor with an indirect bandgap, small atomic number and low density will achieve high efficiency.

Diamond has the highest theoretical efficiency as a betavoltaic battery energy converter, but the development of its experimental devices with a PIN junction is still a challenging task [41]. N-type diamond materials have unacceptable plenty defects due to the immature doping technology, and it is difficult to form a stable and reliable PN junction. The performance of diamond experimental devices is still far below expectations, with the *V_oc_* of PIN-type devices only 0.2–1 V [42,43,44]. In general, SiC will be more suitable for betavoltaic batteries when considering both theoretical efficiency and material preparation maturity.

### 3.2. Prediction of Ultra-Wide-Bandgap Semiconductor

The device efficiency *η*_d_ is related to the EHP creation energy *ε*, energy deposition of beta particles *E(x)* and collection efficiency of EHPs *CE(x)*, which can be summarized as the following equation:(10)$$\begin{array}{c} \eta_{d}=f\left[\varepsilon^{\prime },E\left(x\right),CE\left(x\right)\right] \end{array}$$

Compared with Larry Olsen’s prediction, Equation (10) corrects *ε* as *ε′* based on experimental results and further emphasizes the effect of *E(x)* and *CE(x)*. The characteristics of representative ultra-wide-bandgap semiconductors are shown in Table 2. The *η_d_* changing with different semiconductors under different conditions is shown in Figure 5c. Correcting *ε* to *ε′* based on the black line, the result is shown as the blue line. The yellow and red lines were obtained by further considering of *E(x)* and *E(x)* together with *CE(x)*, respectively. It can be seen that EHP creation energy *ε* playing a decisive role in the trend, i.e., experimental *ε′* is not positively proportional to the *E_g_*, leads to device efficiency *η_d_* not being simply proportional to *E_g_*. The introduction of the energy deposition *E(x)* effect leads to a decrease in *η_d_*, which is more pronounced for a betavoltaic battery using semiconductors with higher atomic numbers and material density, such as GaN and Ga_2_O_3_. The introduction of *CE(x)* further reduces the device efficiency, which is especially pronounced for AlN. The short *L_p_* of AlN results in a low *CE(x)* outside the junction, which makes its *η_d_* even lower than that of SiC. The BN betavoltaic battery has the overall highest ideal efficiency, which means BN will be the optimal candidate among ultra-wide-bandgap semiconductors for betavoltaic batteries, although it is currently challenge to prepare stable homogeneous PN junctions [18].

### 3.3. Efficiency Prediction of the SiC Experimental Device

SiC betavoltaic batteries with a PIN junction were designed and fabricated to compare their performance with other semiconductor betavoltaic batteries, and their parameters were determined according to the energy deposition distribution in Figure 4b. The output performance of the SiC device under an isotope source or electron beam was predicted by the following method, described in Section 2.1.2, as shown in Table 3. The prediction efficiency under a ^63^Ni source is 17.54%, which is obviously lower than the ideal efficiency predicted in Table 1. This is due to the significantly reduced energy deposition in the SiC semiconductor compared to the ideal condition (as mentioned in Figure 4b), as shown in Figure 5d. The energy deposition of beta particles in different parts for the SiC device under a ^63^Ni source is shown in Figure 5d. Energy deposition in the ^63^Ni source and source substrate is up to 60%, which significantly reduces the energy deposition in the SiC semiconductor and makes the predicted efficiency much lower than the ideal value obtained in Table 1.

### 3.4. Measurement of the SiC Experimental Device

#### 3.4.1. Measurement of Electron Beam

SiC betavoltaic battery devices were fabricated, and the I–V characteristics under an electron beam were analyzed. The obtained *I_sc_*, *V_oc_* and *FF* factors are in the range of 141.38 nA to 195.28 nA, 1.23 V to 1.54 V and 0.86 to 0.93, respectively. The best achieved device efficiency is 14.88%, according to Equation (1), with the corresponding I–V characteristics as shown in Figure 6a. This obtained SiC device efficiency is significantly higher than the available GaN device efficiency at 6.6% [9] and diamond device efficiency at 9.4% [8]. It is verified experimentally that SiC is the optimum wide-bandgap semiconductor for the betavoltaic battery.

The measured ideal factor of the best SiC device can be calculated as 1.36, according to the dark current analysis of I–V characteristics. A large ideal factor means a high proportion of recombination current in the junction region, which will lead to the decrease in barrier height and battery efficiency. The ideal factor was assumed to be 1 in the prediction situation, which results in the predicted *V_oc_* and *η* being higher than the measured results, and can be modified based on the measured ideal factor. The modified *V_oc_* and *η* predictions were 1.66 V and 16.32%, respectively, which were close to the measured results, as shown in Table 3. The defects in material growth and device surface treatment will increase the ideal factor, as it relates to the recombination current. Surface oxidation and the roughness of the electrode caused by the annealing process may be the reason why the ideal factor is higher than 1. More detailed surface treatment process exploration will be carried out in the future to further improve the battery performance.

#### 3.4.2. Measurement of ^63^Ni Source

The I–V characteristics of SiC experimental devices with a ^63^Ni source were also analyzed. The obtained *I_sc_*, *V_oc_* and *FF* factors are in the range of 2.85 nA to 2.91 nA, 1.22 V to 1.41 V and 0.81 to 0.89, respectively. The best achieved device efficiency is 7.31%, which is lower than that under an electron beam, as shown in Table 3, with the corresponding I–V characteristics as shown in Figure 6b. This is due to the self-absorption effect of the isotope source, which results in lower energy deposition in a semiconductor than an electron beam.

The measured short-circuit current *I_sc_* of the SiC betavoltaic battery devices prepared in this work is slightly lower than the theoretically predicted value of 2.99 nA, with an overall difference of less than 3%. The measured value of the fill factor *FF* is similar to the theoretically predicted value of 0.91, whereas the measured value of the open-circuit voltage *V_oc_* has a higher difference from the theoretically predicted value of 1.89 V, resulting in a certain discrepancy in the energy conversion efficiency. According to Equation (6), the open-circuit voltage *V_oc_* increases with the decrease in the reverse saturation current *I*_0_. Substituting the measured reverse saturation current *I*_0_ value of 4.26 × 10^−26^ A into Equation (6), the corrected open-circuit voltage is obtained as 1.42 V, which is close to the measured result, and the theoretical device conversion efficiency is 12.83%, as shown in Table 3.

The output performance of the SiC betavoltaic batteries prepared in this paper is compared with the experimental results of other SiC betavoltaic batteries, and the results are shown in Table 4. The results show that the SiC betavoltaic batteries in this paper have a high fill factor and energy conversion efficiency.

## 4. Conclusions

In order to screen the optimum selection of wide-bandgap semiconductors, the device efficiencies of betavoltaic batteries with different semiconductors were analyzed by considering principle factors in the generation and collection processes of EHPs. For typical wide-bandgap semiconductors, the prediction efficiency of betavoltaic batteries with GaN was lower than that of SiC and diamond in this work. And the main reasons are summarized as follows: (1) The experimentally determined EHP creation energy deviated from the simple empirical assumption due to the complex band structure of wide-bandgap semiconductor materials. (2) The EHPs collection efficiency of GaN is relatively lower because of the short lifetime of its minority carrier as a direct-gap semiconductor, and shorter diffusion length resulting from more apparent defects in its heteroepitaxial growth process. (3) GaN has a higher atomic number and density than SiC and diamond, leading to higher backscattering and more energy loss of beta particles. The ideal efficiencies of betavoltaic batteries with ultra-wide-bandgap semiconductors were further predicted. Similarly, the ideal efficiencies of Ga_2_O_3_ and AlN betavoltaic batteries are even relatively lower than that of the SiC betavoltaic battery, while the BN betavoltaic battery has the potential to achieve the best performance. Considering both the theoretical efficiency prediction and the preparation maturity, SiC is the currently optimum choice for betavoltaic batteries, and its superiority was verified through experiments. The best obtained device efficiencies of SiC devices under an electron beam and a ^63^Ni source are 14.88% and 7.31%, respectively, in this work, and the energy conversion efficiency reaches 2.34~2.56%, an increase of about one time, which is obviously higher than that of betavoltaic batteries with other semiconductor materials reported in the literature. The device efficiency of SiC betavoltaic batteries can be further improved by optimizing the surface process to reduce the device ideal factor. The performances of betavoltaic batteries are worthy, as expected, along with the rapid development of preparation techniques related to ultra-wide-bandgap semiconductors. This work provides theoretical support for the development of betavoltaic batteries with ultra-wide-bandgap semiconductors.

## Figures and Tables

**Figure 2 nanomaterials-15-00635-f002:**
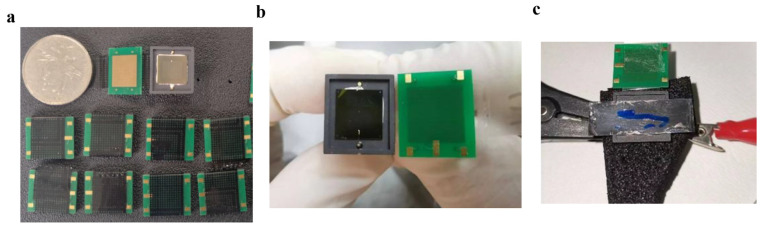
Physical diagram of the devices. (**a**) Series of physical samples of betavoltaic batteries prepared. (**b**) SiC-based betavoltaic battery. (**c**) ^63^Ni isotope source loading test.

**Figure 4 nanomaterials-15-00635-f004:**
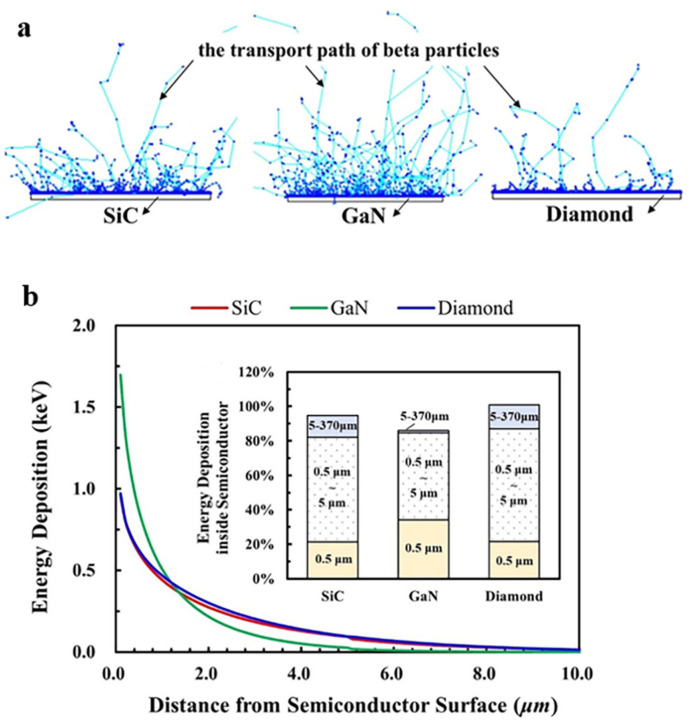
Effects of the energy deposition of beta particles. (**a**) The backscattering of different semiconductors under 1000 beta particles. (**b**) The energy deposition distribution of ^63^Ni decaying beta particles in each semiconductor.

**Figure 5 nanomaterials-15-00635-f005:**
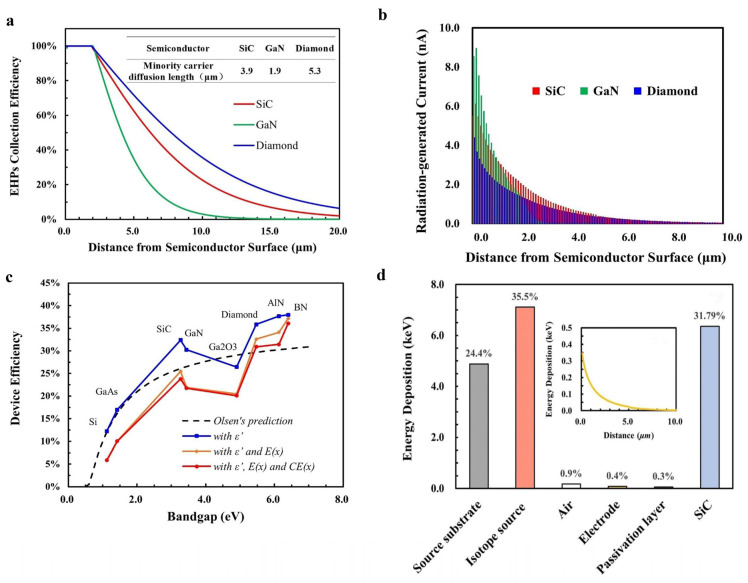
Predictions for different bandgap semiconductors. (**a**) The EHPs collection efficiency of different semiconductors. (**b**) The radiation-generated current distribution of different betavoltaic batteries. (**c**) The device efficiency of a betavoltaic battery using different semiconductors by considering different effects. (**d**) The energy deposition distribution of beta particles in SiC semiconductor under a ^63^Ni volume source. Inset: the energy deposition of β particles in different parts for the SiC betavoltaic battery device under a ^63^Ni source.

**Figure 6 nanomaterials-15-00635-f006:**
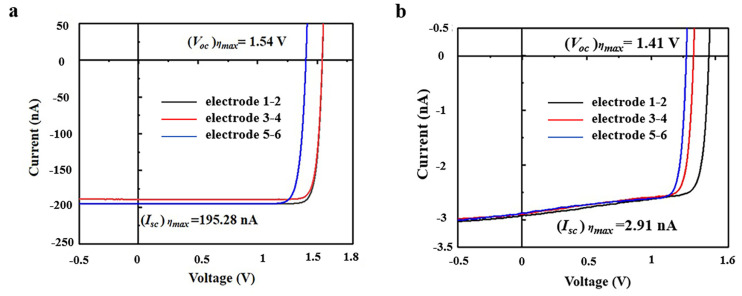
Measurement of the devices. (**a**) The I-V characteristics of SiC betavoltaic battery device under an electron beam at 20 kV acceleration voltage. (**b**) The I-V characteristics of the SiC betavoltaic battery device under a ^63^Ni isotope source.

**Table 1 nanomaterials-15-00635-t001:** The ideal device efficiency of betavoltaic battery using typical wide- and ultra-wide-bandgap semiconductors.

Semiconductor	SiC	GaN	Diamond	Ga_2_O_3_	AlN	BN
*J_sc_*/nA·cm^−2^	110.16	95.18	72.18	51.18	43.62	64.67
*V_oc_/*V	2.24	2.40	4.39	3.81	5.31	5.75
*η_d_*	22.17%	20.69%	29.29%	17.98%	21.57%	34.65%

**Table 2 nanomaterials-15-00635-t002:** The characteristics of representative ultra-wide-bandgap semiconductors [36,44,45,46,47,48] **.

Semiconductor	Ga_2_O_3_	AlN	BN
Bandgap/eV	4.9	6.13	6.4
Density/g·cm^−3^	6.1	3.26	3.49
Electron–hole pair creation energy/eV	15.6	14.4 *	15.0 *
Relative dielectric constant	10	8.6	7.1
Intrinsic carrier concentration/cm^−3^	2.81 × 10^−23^	3.82 × 10^−36^	8.83 × 10^−39^
Minority carrier diffusion length/μm	0.325	0.15	7.6
Carrier mobility/cm^2^·V^−1^yggs^−1^	20	14	500
Bandgap type	direct	direct	indirect

* It is estimated based on the relation (2.15Eg+1.23), which is re-extracted from experimental data from existing materials. ** The SiC, Ga2O3 and BN mentioned in this paper are 4H-SiC, β-Ga2O3 and h-BN respectively.

**Table 3 nanomaterials-15-00635-t003:** Prediction and measurement results of SiC betavoltaic battery devices.

Source	Condition	*I_sc_/*nA	*Voc/*V	*FF*	*η_d_*	n
Electron beam	original prediction	197.71	2.25	0.94	23.17%	1
	prediction with experimental n	197.71	1.66	0.92	16.32%	1.36
	measurement	195.28	1.54	0.89	14.88%	1.36
^63^Ni	original prediction	2.99	1.89	0.91	17.54%	1
	prediction with experimental *I*_0_	2.99	1.42	0.88	12.83%	1.51
	measurement	2.91	1.41	0.81	7.31%	1.51

**Table 4 nanomaterials-15-00635-t004:** Output performance of different betavoltaic batteries.

Semiconductor	*Voc/*V	*FF*	*η*	References
SiC	0.50	0.30	1.3%	[49]
	0.72	0.51	1.2%	[50]
GaN	0.14	0.14	1.60%	[51]
	1.62	0.55	1.13%	[52]
Diamond	1.02	0.70	1.25%	[15]
This work (SiC)	1.22–1.41	0.81–0.89	2.34–2.56%	--

## Data Availability

The raw data supporting the conclusions of this article will be made available by the authors on request.

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
