# Peer review of "Optimal Selection and Experimental Verification of Wide-Bandgap Semiconductor for Betavoltaic Battery"

_nanomaterials, 2025, doi:10.3390/nano15090635_

Round 1
Reviewer 1 Report
Comments and Suggestions for Authors
The manuscript deals with the selection and experimental results regarding of wide
bandgap semiconductor for betavoltaic battery. The manuscript is convinging and interesting for many applications.
Author Response
Dear reviewer
Thank you for your thoughtful review of our manuscript and for your positive feedback on the importance and applicability of our work on wide bandgap semiconductors for betavoltaic batteries. We are pleased that you found our experimental results convincing and that our research topic has potential applications.
Your recognition of the value of the manuscript is very encouraging to our team. We have carefully considered your feedback to ensure that the manuscript maintains rigorous scientific accuracy while clearly communicating the results. We would like to thank you for your time and constructive feedback.
We shall look forward to hearing from you at your earliest convenience.
Yours sincerely,
Jiachen Zhang
E-mail: zhjiachen.good@foxmail.com
Reviewer 2 Report
Comments and Suggestions for Authors
The authors studied the promising materials as betavoltaic device. The authors found that the SiC had higher performance than the other materials. The conversion efficiency was =2.5 %. This performance is very exciting. Furthermore, the Ni source and the electron beam enhanced the performance. This information is helpful for the readers. However, there are some concerns about the introduction and the result sections. If the authors appropriately revise the manuscript, this study will meet the criteria for the publication in Nanomaterials.
Comment list
Comment 1: Please describe the mechanism of energy transfer from beta particle to electron.
Comment 2: Is the electrode choice important for the performance?
Comment 3: What determines the efficiency of electron generation?
Comment 4: The wide bandgap semiconductors are used in various applications. The authors should comment on the important studies in various fields. For example, there are important studies using semiconductors: thermoelectric material (Appl. Phys. Lett. 118, 151601 (2021).); piezoelectric material (Adv. Funct. Mater. 21, 628 (2011).). The authors cannot ignore such studies. To attract broad interest, the authors revise the manuscript.
Author Response
Dear reviewer
Thank you for your thorough evaluation of our manuscript and your encouraging comments on the potential of SiC-based betavoltaic batteries. These comments are insightful and allow us to improve the quality of the manuscript. Below is our point-by-point response to each of your comments. Changes in the text are highlighted in different colours. Once again, thank you for your constructive comments. We believe that these revisions have improved the rigour and completeness of the manuscript.
We shall look forward to hearing from you at your earliest convenience.
Yours sincerely,
Jiachen Zhang
E-mail: zhjiachen.good@foxmail.com
Comment 1: Please describe the mechanism of energy transfer from beta particle to electron.
Response 1: Thanks for your suggestion. The description of the energy transfer mechanism from beta particles to electrons consists of the following steps: beta particles decay from an isotope source, some of which are emitted from the surface of the source, pass through the electrodes and are injected into the semiconductor material; electron-hole pairs (EHPs) are produced, which are then separated by the action of a built-in electric field in a semiconductor junction, such as a Schottky or PN junction; and finally the electrons and holes are collected by the electrodes and a current is produced. This is also available in revision p1.
Comment 2: Is the electrode choice important for the performance?
Response 2: Thank you for your suggestion. Yes, it is important for electrode selection. Based on our previous research results, Ni/Ti/Al/Ni metal was finally selected as the ohmic contact electrode metal for SiC-based betavoltaic batteries. By adding Ni protective metal, the oxidation problem on the sample surface during the test process was improved and the homogeneity of the measurement results was better. At the same time, in order to investigate the effect of different metals on the output performance of the irradiated nuclear battery, the Ti/Al/Au ohmic contact electrode of the PN junction SiC-based betavoltaic battery of Xi'an University of Electronic Science and Technology was selected as the control group and compared with the Ni/Ti/Al/Ni metal electrode preferred in this paper, with the metal electrode thicknesses of 50/100/100 nm and 50/30/90/50 nm, and the annealing temperatures of 50/30/90/50/50 nm. The metal electrode thicknesses were 50/100/100 nm versus 50/30/90/50, and the annealing temperatures were all 950 °C and annealing times were 5 min. Compared to Ti/Au, the Ni/Ti/Al/Ni metal electrode samples showed lower open circuit voltages Voc and higher short circuit currents Isc, and devices had a higher FF and Pmax. The specific effects of the different electrodes can be seen in the table below.
|
|
Sample |
Isc (nA) |
Voc (V) |
FF |
Pmax(nW) |
Æž |
|
Ti/Al/Au |
1 |
1.66 |
1.77 |
0.53 |
1.55 |
1.21% |
|
2 |
1.12 |
1.76 |
0.66 |
1.30 |
1.01% |
|
|
3 |
1.13 |
1.78 |
0.65 |
1.32 |
1.03% |
|
|
4 |
1.34 |
1.82 |
0.69 |
1.70 |
1.33% |
|
|
Ni/Ti/Al/Ni |
1 |
2.89 |
1.41 |
0.81 |
3.28 |
2.56% |
|
2 |
2.88 |
1.22 |
0.89 |
3.11 |
2.43% |
|
|
3 |
2.85 |
1.37 |
0.86 |
3.36 |
2.63% |
|
|
4 |
2.91 |
1.31 |
0.82 |
3.12 |
2.44% |
Comment 3: What determines the efficiency of electron generation?
Response 3: Thank you for your suggestion. The efficiency of electron generation, denoted G(x), is a measure of the number of electron-hole pairs generated at different positions. It is closely related to the energy deposition of the decaying particles in the semiconductor, the activity of the isotopic source, and the average ionisation energy, and can be expressed as follows:
G(x)= AE(x)/ε
where ε is the average ionisation energy, also known as the electron-hole pair generation energy, reflecting the energy required to produce a single electron-hole pair, a characteristic of semiconductor materials; A is the activity of the isotopic source; E(x) is the energy deposition function of decay particles in semiconductors, which needs to be simulated by Monte Carlo methods, and the related simulation methods and modelling are described in this paper.
Comment 4: The wide bandgap semiconductors are used in various applications. The authors should comment on the important studies in various fields. For example, there are important studies using semiconductors: thermoelectric material (Appl. Phys. Lett. 118, 151601 (2021).); piezoelectric material (Adv. Funct. Mater. 21, 628 (2011).). The authors cannot ignore such studies. To attract broad interest, the authors revise the manuscript.
Response 4: Thank you for your suggestion. We have added advances in wide-bandgap semiconductor research in thermoelectric materials, piezoelectric materials, and other key areas on page 2, pp. 86-88 of the revised manuscript, with citations to key literature, including Appl. Phys. Lett. 118,151601 (2021) and Adv. Funct Mater. 21,628 (2011), which were mentioned by the reviewers. We thank you for your suggestion, which we believe will greatly improve the depth and completeness of the article.
Reviewer 3 Report
Comments and Suggestions for Authors
This reviewer went through the submitted work entitled “Optimal Selection and Experimental Verification of Wide Bandgap Semiconductor for Betavoltaic Battery” to Nanomaterials/MDPI by J. Zhang et al. The use of wide bandgap semiconductors for the conversion material of radioisotope microbatteries offers a variety of advantages compared to using Si. The energy conversion efficiency may be increased using a wide bandgap material. Silicon carbide (SiC), with a bandgap of 3.23 eV for its 4H-polytype, is one of the low-cost candidate materials that has been applied in the area of ionizing radiation detection (e.g., X-rays, neutrons, and charged particles), attributed to its excellent radiation tolerance and high thermal stability. The optimal choice of material is a significant area of research. The authors have made a comprehensive analysis in understanding the effects of different semiconductor characteristics on the battery energy conversion process. Several materials are analysed and presented. The manuscript is well organized and can bring some new insights to the field.
The following points require some clarity:
- The title of the work is unclear; reconsider the wordings and make it effective.
- Maybe a good idea to include the list of selected/reported semiconductors for this application, with the present state-of-the-art status
- What is the semiconductor conversion efficiency (ηS) in Shockley-Queisser approximation?
- Did the authors determine the diffusion length and leakage current?
- In the estimation of e-h pair generation rate, do the authors consider the factors such as top layer thickness, junction depth, doping concentrations, etc.?
- The relevant articles published recently on the charge carriers transport characteristics associated with the numerical calculations can be included and referred to the papers such as doi.org/10.3390/electronics13214223; and doi.org/10.3390/coatings13091657
- The evaluation of the beta cell and coupling efficiencies must be clearly emphasized at both the results and discussions, along with the conclusion section.
- Comment on the prediction of radiation damage.
- The inset in Figure 4d is not readily visible.
Author Response
Dear reviewer
Thanks for your letter and comments concerning the manuscript entitled “Optimal Selection and Experimental Verification of Wide Bandgap Semiconductor for Betavoltaic Battery”. The comments were insightful and enabled us to improve the quality of our manuscript. The following pages are our point-by-point responses to each of the comments. Revisions in the text are shown using fonts of different colors for additions. We believe that these revisions have improved the rigour and completeness of the manuscript.
We shall look forward to hearing from you at your earliest convenience.
Yours sincerely,
Jiachen Zhang
E-mail: zhjiachen.good@foxmail.com
Comment 1: The title of the work is unclear; reconsider the wordings and make it effective.
Response 1: Thank you for your comments. The core of this work lies in the optimal selection and experimental validation of wide bandgap semiconductors for betavoltaic batteries. After a thorough evaluation, we have decided to retain the title in line with our core research objectives.
Comment 2: Maybe a good idea to include the list of selected/reported semiconductors for this application, with the present state-of-the-art status.
Response 2: Thank you for your comments. The article compares the current state-of-the-art efficiencies of a number of broadband semiconductor materials, including SiC, GaN, diamond, Ga2O3, AlN and BN, which can be found in Tables 1 and 2 of the article.
Comment 3: What is the semiconductor conversion efficiency (ηS) in Shockley-Queisser approximation?
Response 3: Thank you for your comments. The black curves in Figures 2-a and 4-c are Olsen's prediction, which is ηS in the limiting case of the Shockley-Queisser approximation, and also using Eq. 8 we get.
Comment 4: Did the authors determine the diffusion length and leakage current?
Response 4: Thank you for your comments. The diffusion lengths of semiconductor materials are obtained from literature searches. The diffusion lengths of different semiconductor materials are given in section 3.1.3 of the article, where the diffusion lengths are 1.9 μm for GaN, 3.9 μm for SiC and 5.3 μm for diamond. The magnitude of the leakage current is indicative of the level of dark current in the device, as expressed in equation :
I0=Sqni [(Dpni)/(LpND ) exp(qV/kT)+w/(2τp ) exp(qV/2kT)]
where S represents the cross-sectional area of the depletion region in cm2; Dp and Dn denote the minority carrier diffusion coefficients of the P-type and N-type regions, respectively, in cm2/s; and Lp and Ln are the minority carrier diffusion lengths of the P-type and N-type regions, respectively, in cm. Subsequently, a variety of optimisation experiments were carried out on the samples to stabilise the leakage current at levels ranging from 1.32 × 10-25 A to 2.44 × 10-24 A.
Comment 5: In the estimation of e-h pair generation rate, do the authors consider the factors such as top layer thickness, junction depth, doping concentrations, etc.?
Response 5: Thank you for your comments. Yes, we have chosen to use this model that you have replied to. The investigation is chiefly concerned with the effect of material parameters, and thus all models are constructed from the same structural parameters. However, your suggestion is very good, and in the future we will study the top layer thickness, junction depth, doping concentration and other factors for each different material.
Comment 6: The relevant articles published recently on the charge carriers transport characteristics associated with the numerical calculations can be included and referred to the papers such as doi.org/10.3390/electronics13214223; and doi.org/10.3390/coatings13091657
Response 6: Thank you for your comments. After carefully reading the comments mentioned, we have added relevant content in section 3.1.3 of the article and cited research papers on carrier transport properties related to numerical calculations. We are grateful for your suggestions, which in our opinion will greatly improve the depth and completeness of the article.
Comment 7: The evaluation of the beta cell and coupling efficiencies must be clearly emphasized at both the results and discussions, along with the conclusion section.
Response 7: Thank you for your comments. Regarding the evaluation of beta cells and coupling efficiencies, we have added and discussed them in the results of the revised manuscript. Thank you for your suggestions. We believe they will greatly improve the depth and completeness of the article.
Comment 8: Comment on the prediction of radiation damage.
Response 8: Thank you for your suggestion, the wide bandwidth material we selected is very little affected by irradiation damage and therefore is not discussed in detail in this paper. However, we have evaluated the long term operational stability and have explored the effect of irradiation damage. The magnitude of variation in the output performance of SiC-based betavoltaic batteries for different equivalent operating hours is shown in the table below.
|
Equivalent activity (Ci) |
Equivalent operating hours (year) |
FF |
Voc(V) |
|
17.5 |
0.5 |
-1.1% |
0% |
|
2 |
0% |
+0.8% |
|
|
5 |
+1.1% |
+2.3% |
|
|
10
|
2 |
-2.9% |
+3.8% |
|
4 |
-5.8% |
+0.8% |
|
|
6 |
-8.7% |
-0.8% |
|
|
8 |
-4.3% |
+3.1% |
Comment 9: The inset in Figure 4d is not readily visible.
Response 9: We thank the reviewers for their comments. We are sorry for these mistakes. We have already made corrections to Figure 4d in the manuscript. We have also modified Figures 1-4 to make them clearer.
Reviewer 4 Report
Comments and Suggestions for Authors
The paper concerns using wide bandgap semiconductors for betavoltaic batteries. The focus is on predicting the betavoltaic batteries' performance characteristics affected by the specifics of selected wide bandgap semiconductors. Especially, the paper addresses the inconsistency of the Larry Olsen’s model predictions as demonstrated by the SiC and GaN betavoltaic batteries performance. The work is novel with high practical relevance.
The main criticism we have is regarding the paper organization. In general, the paper format is OK, however, the mix-up of simulated results with experimental data is confusing to a reader.
In section 2.2.1 the authors provide experimental fabrication details, however, there is no info on the real material (SiC?) used for fabrication of the tested devices. For example, what type of SiC was used, deposition method, doping level, ect. What metals were used for Ohmic contact? Show the tested prototype device picture, a schematic if different from Fig.1(d). We recommend reorganizing the manuscript and making it clear what the real devices tested are.
It is not clear how the measurement of Ni source was performed. Show the picture of a complete betavoltaic battery tested here with the Ni source included.
Fig.1(c) needs further explanation, what is the meaning of different colors in the figure?
It is well-known that the quality of GaN, AlN, Ga2O3, and BN materials depends on the selected growth method. In section 3.1.1 the authors indicated briefly when considering the EHP collection efficiency. Unfortunately, the EHP collection efficiency in any of these hosts depends strongly on the thin film morphology, which defines the defect density. How was this issue considered in the simulations? It would be valuable to include e.g. specific defect density of GaN epilayers for typical MOCVD, CVD, HVPE etc. growth methods and show how the affected EHP collection efficiency changes the betavoltaic battery performance.
How do the simulation results of a segmented semiconductor with 450 layers, with a thickness of 0.1 μm relate to the betavoltaic battery experimentally tested?
Provide the electron beam current density in the electron irradiation experiment. Show the electron penetration depth profile for the simulated structure.
Regarding the BN for betavoltaic battery, what BN structure was considered in the simulation (h-BN or c-BN)?
It is known that high hole concentrations in ultra-wide bandgap semiconductors like AlN and BN are difficult to achieve. Comment on the ideal device efficiency of betavoltaic battery in Table 2 considering the realistic high hole concentrations achievable in these materials.
It would be valuable to consider the effect of structural defects in GaN, AlN, Ga2O3, and BN on the ideal efficiency. Was the SHR model considered in the simulation? If not, why?
How was the Bertuccio-Maiocchi-Barnett (BMB) relationship considered here Ref. 23?
In conclusion, the paper idea is novel and reports on valuable observations relevant to contemporary research efforts in energy and battery fields, however, the manuscript suffers from missing experimental data, missing theoretical information, and poor organization. We recommend major revision before further considerations for publication.
Comments on the Quality of English LanguageNone
Round 2
Reviewer 2 Report
Comments and Suggestions for Authors
Everything was cleared. This study is worth publishing.
Author Response
Dear Reviewer,
We would like to express our gratitude for your comprehensive evaluation of our manuscript and the positive feedback you have provided. We are greatly encouraged by your affirmation that "this study is worthy of publication" and that all concerns have been adequately addressed. Your scholarly insights throughout the review process have significantly strengthened the rigor and clarity of our work. We sincerely appreciate your time and expertise in assessing our research, and we are honoured by your endorsement for publication.
Yours sincerely,
Jiachen Zhang
E-mail: zhjiachen.good@foxmail.com
Reviewer 3 Report
Comments and Suggestions for Authors
The changes embedded in the revised version appear to be satisfactory. Therefore, in this reviewer's opinion, the work can be published.
Author Response
Dear Reviewer,
We are immensely grateful for your acknowledgement that "the changes embedded in the revised version appear to be satisfactory" and greatly value your concluding assessment that "the work can be published." Your discerning observations during the review process have been instrumental in refining both the technical precision and scholarly presentation of this research. We would like to express our profound gratitude for the time and intellectual rigour you have dedicated to improving this manuscript. We are honoured by your endorsement of its publication.
Yours sincerely,
Jiachen Zhang
E-mail: zhjiachen.good@foxmail.com
Reviewer 4 Report
Comments and Suggestions for Authors
This is a revised manuscript. Our comments are shown below.
Response to Comment 1 Author’s reply: Show a single Ohmic contact I/V characteristic measured. We recommend showing the picture of the prototype devices as a fig. insert in the manuscript or a new figure in the Supplementary. The contact info is critical because, as the authors have stated, “Considering the difficulty of achieving high doping in the materials and the difficulty of achieving ohmic contacts”.
Response to Comment 2 Author’s reply: Thank you for the additional info provided. We recommend adding the content of Comment 2 to the text or including it in the Supplementary.
Response to Comment 3 Author’s reply: Please revise the Fig.1(c) caption by adding the explanation “The colors in the figure are used to highlight the layering.”
Response to Comment 4 Author’s reply: If the authors decide not to discuss the EHP collection efficiency affected by the film morphology, we recommend indicating in the discussion what would be over- or under-estimated error resulting from considering the idealized model/material. A simple approach to include the SiC host defect types and density would be a good reference point for a reader to assess the reported EHP collection efficiency. Please revise the text.
Response to Comment 5 Author’s reply: Unfortunately, we did not find the authors’ reply satisfactory. We recommend that authors compare the proposed simulated model shown in Fig.1(c) with a real device structure tested (Fig.1(d) (?)) and then discuss the differences. For example, why does the model include 450 layers, whereas a tested device has a different structure?
Response to Comment 6 Author’s reply: Thank you for the info. The electron beam density must be expressed per surface area because it is more informative and relevant to the discussion than the electron beam intensity.
Response to Comment 7 Author’s reply: Thank you for the info, however, the h-BN was not mentioned in the text/tables. Please revise.
Response to Comment 8 Author’s reply: The rebuttal provided is generic and does not bring any new information. We recommend that authors estimate the upper limit efficiency uncertainty in Table 2. This does not call for new experiments but more insightful analysis of the available data.
Response to Comment 9 Author’s reply: The rebuttal provided is generic and does not bring any new information. The future study plans are OK, but we are considering the current manuscript. We recommend that authors address/estimate how the defects in real materials/devices will affect the reported efficiency/performance of the reported betavoltaic batteries' performance. It will only add to the quality of the paper, making its impact much more significant.
Response to Comment 10 Author’s reply: The content of the author’s rebuttal should be included in the manuscript narrative.
We appreciate the author’s willingness to revise the manuscript and recommend a second round of revisions to address the indicated deficiencies.
Author Response
Dear Reviewer
Thank you for your letter and comments on the manuscript. These comments are insightful and allow us to improve the quality of the manuscript. Below is our point-by-point response to the comments. Changes in the text are highlighted in different colours. Once again, thank you for your constructive comments. We believe that these changes have improved the rigour and integrity of the manuscript.
Sincerely
Jiachen Zhang
E-mail: zhjiachen.good@foxmail.com

Round 3
Reviewer 4 Report
Comments and Suggestions for Authors
This is a 2nd revision. The authors addressed the reviewer’s concerns adequately. Thank you.